# Fabrication of High-Performance CNT Reinforced Polymer Composite for Additive Manufacturing by Phase Inversion Technique

**DOI:** 10.3390/polym13224007

**Published:** 2021-11-19

**Authors:** Pooyan Parnian, Alberto D’Amore

**Affiliations:** Department of Engineering, University of Campania “Luigi Vanvitelli”, Via Roma 29, 81031 Aversa, Italy; pooyan.parnian@unicampania.it

**Keywords:** Additive Manufacturing, nanocomposite, high-performance polymer, Carbon nanotubes, mechanical properties, thermal properties

## Abstract

Additive Manufacturing (AM) of polymer composites has enabled the fabrication of highly customized parts with notably mechanical properties, thermal and electrical conductivity compared to un-reinforced polymers. Employing the reinforcements was a key factor in improving the properties of polymers after being 3D printed. However, almost all the existing 3D printing methods could make the most of disparate fiber reinforcement techniques, the fused filament fabrication (FFF) method is reviewed in this study to better understand its flexibility to employ for the proposed novel method. Carbon nanotubes (CNTs) as a desirable reinforcement have a great potential to improve the mechanical, thermal, and electrical properties of 3D printed polymers. Several functionalization approaches for the preparation of CNT reinforced composites are discussed in this study. However, due to the non-uniform distribution and direction of reinforcements, the properties of the resulted specimen do not change as theoretically expected. Based on the phase inversion method, this paper proposes a novel technique to produce CNT-reinforced filaments to simultaneously increase the mechanical, thermal, and electrical properties. A homogeneous CNT dispersion in a dilute polymer solution is first obtained by sonication techniques. Then, the CNT/polymer filaments with the desired CNT content can be obtained by extracting the polymer’s solvent. Furthermore, optimizing the filament draw ratio can result in a reasonable CNT orientation along the filament stretching direction.

## 1. Introduction

Additive Manufacturing (AM) is the fabrication process of materials incrementally and layer-by-layer from 3D models known as 3D printing. During more than 20 years of development of AM process, a wide range of industries has affected such as automotive, aerospace, architectural design, digital art, biomedical, and so on [1]. In recent years, significant progress has been made in AM technology due to its adaptability and economical aspects regarding traditional manufacturing techniques, especially for rapid prototyping applications. [2].

Based on the ASTM Standards (ASTM 2012), AM operations can be classified into seven principal categories, namely: binder jetting, directed energy deposition, material extrusion (FFF or FDM), material jetting, powder bed fusion (SLS, SLM), sheet lamination (LOM), and vat photopolymerization (SLA) [3]. Among many operations of AM, the FFF (fused filament fabrication) method has gained meaningful attention because of its advantages over other operations: ease of access and use, inexpensive feed materials, and rather higher quality of fabricated parts. The FFF is an extrusion-based process. The final component is produced by superposing layers of extruded filaments [4,5].

Despite the plentiful merits of FFF over traditional manufacturing techniques, the FFF printed parts often suffer from a lack of mechanical properties and their application in final load-bearing products. This problem could be attributed to the nature of the used material. However, many recent articles have addressed this limitation of the FFF process through attempts to solve it by optimization the printing parameters. In addition, the printed part itself has some constraints due to insufficient adhesion between subsequent layers and internal porosities. Accordingly, a printout with 100% filling is almost impossible [4,6].

Utilizing reinforced polymers or composite materials as a filament to feed into the FFF printer is an action to alleviate the deterioration in mechanical properties. In composite materials, the matrix is a functional phase. It is responsible for the rheological and mechanical properties of the whole composite by supporting the existent reinforcements. Generally, the surface of glass and carbon fibers [7,8,9] is chemically modified to enhance the mechanical properties of FFF printed parts. Therefore, the employment of a proper matrix and reinforcement may cause an improvement in the mechanical properties and the achievement of other properties such as high thermal and electrical conductivity [4].

Theoretically, Carbon nanotubes (CNT) are among the most used reinforcements when high strength, lightweight, and high-performance composite is needed. Composites with CNT reinforcements are widely used in aircraft, automobiles, and sports gear. CNTs with a high modulus of elasticity greater than 1 TPa (the elastic modulus of a diamond is 1.2 TPa) have a 10–100 times higher strength than steels [10]. Moreover, CNTs with the exceptional thermal stability of around 2800 °C in a vacuum and an electrical conductivity of about 103 S/cm. High conductivity can be attained at a very low CNTs’ concentration from 0.0025 up to 4 wt.%, because of their high length, L, to diameter, D, ratio (L/D), ranging from hundreds to 10,000 [11].

The excellence mentioned above of CNTs resulted in their widespread application as a reinforcement and filler in composites to boost the produced Nanocomposite’s mechanical, thermal, and electrical properties. Furthermore, CNTs are used in different fields, including batteries [12], solar cells [13], microwave absorption [14], natural fiber composites [15], chemical sensors [16], corrosion protection, and adsorbents [11,17].

Alongside the previous unique features of CNTs, there are some difficulties to exerting CNTs in frequent volatile organic solvents or polymeric matrices due to their strong π–π interactions and Van del Walls forces between pristine CNTs. Such actions lead to the agglomeration of nanotubes in the polymer matrices and decrease their dispersion properties significantly [18]. Moreover, a critical factor in the dispersion of the nano-sized CNTs into polymeric matrices is their physical nature. It has been demonstrated that the accumulation and agglomeration are the main reason for the deterioration of the resulting composite’s electrical and mechanical properties compared to the theoretical predictions for individual CNTs [19]. In other words, finding an appropriate technique to separate the bundles of CNTs during and after solving them into polymer matrices is a significant concern. It has been proved that surface modification of CNTs, such as chemical modification of the sidewalls, could improve their solubility and dispersion in solvents or polymers, but it has remained a challenge [11,20]

Recent studies have shown a good dispersion of inorganic nanotubes such as boron nitride nanotubes (BNNTs), halloysite nanotubes (HNTs), and imogolite nano-tubes (INTs) in either aqueous or non-aqueous solutions. Although, BNNTs have shown acceptable mechanical properties, in this study CNTs have been investigated due to their high thermal and electrical conductivity at the same value of mechanical properties with a low amount of CNT [21,22]. In order to fabricate CNT composite materials and filaments, it needs a series of actions to be performed in advance such as CNTs modification techniques and polymer solution preparations. On this basis, in the following, some paradigms of recent CNT composite fabrications are explained. The focus is on the improvement of mechanical, thermal, and electrical properties for either 3D printed or CNT nanocomposites. In addition, the effect of different polymer solutions or polymer constituents or even procedures for CNT dispersion on the final properties of a composite system is discussed.

## 2. Current Methods of CNT Nanocomposite Formation

### 2.1. CNT’s Modifications

The production of a homogenous dispersion of CNTs in the polymer solutions needs an operation of filler preparation. By changing the properties of fillers, the interactions between CNTs decreased and prevented their agglomeration. There have been various methods in the literature to overcome the interaction force between CNTs which a common suggestion is the grafting of nanotubes [23]. Accordingly, Shaffer and Koziol researched the synthesis of individual, polystyrene-grafted nanotubes by an in situ radical polymerization technique [24].

In this study, they homemade multi walls carbon nanotubes (MWCNT) with aligned nanotubes and a controlled length and diameter [10]. First, these nanotubes were dispersed in water by treating a 3 + 1 concentrated sulfuric-nitric acid mixture at 45 °C and subsequent washing and following a modification [25]. The dispersion was then combined with a purified styrene monomer and radical initiator (benzoyl peroxide or potassium persulfate). Next, the immiscible layers were agitated rapidly to produce an emulsion, and the polymerization operated under argon at 80 °C for 2 h. After the operation, the organic layer turned black due to the movement of nanotubes from the aqueous layer. Finally, the mixture was diluted with toluene, and the organic layer was centrifuged to deposit the grafted nanotube product [24].

The results represented that the grafted nanotubes immediately dispersed in solvents such as toluene, chloroform, and THF but not in the water or acetone (Figure 1a). The SEM and TEM images of microstructure demonstrated a grafting ratio of 50–90% based on the primary concentration of nanotubes (Figure 1b). The grafting ratio calculated and for the benzoyl peroxide system was approximately 0.5%. Nevertheless, it reached 18% for the potassium persulfate initiator. The results of Shaffer and Koziol showed an adequate dispersion of the grafted CNTs in toluene, chloroform, and THF, which is the first step in acquiring a high-performance nanocomposite.

Lately, a new surfactant has been introduced that can enhance the dispersion of CNTs in polymer composites called Ionic Liquid (IL). It consists of asymmetrical organic cation and organic/inorganic anion. Due to the low volatility, non-flammability, thermal and chemical stability, and large liquid range of IL, it has been broadly expressed as a “green solvent.” Hence, it has become popular in chemistry [14]. CNTs dispersed in IL are usually applied in sensors, actuators, and electrochemistry [26,27]. To this end, Yuan et al. investigated the fabrication of sodium polyacrylate (PAA)/IL-functionalized MWCNT nanocomposites by exploiting an easy solution-casting method. [26].

Chemical oxidation of pristine MWNTs was employed to prepare the carboxylic acid group- functionalized MWCNTs (CNT-COOH). For this purpose, pristine MWCNT (3.0 g), HNO_3_ (65%, 30 mL), and H_2_SO_4_ (98%, 90 mL) were added into a flask with a condenser. The container was submerged in an ultrasonic bath (40 kHz) at 55 °C for 4 h. after chilling to room temperature. The reaction blend was diluted with 4 L of deionized water and put for 24 h to form precipitation of suspended solids at the bottom of the flask. Then, the precipitate was washed and filtered with distilled water by using a Polyvinylidene fluoride (PVDF) membrane of pore size 0.22 μm to neutrality. The carboxylic acid-functionalized MWCNTs (CNT-COOH) formed by drying the filtered solid at 50 °C for 24 h as shown in Figure 2. The schematic of imidazole-functionalized MWCNTs (CNTMi), is shown in Figure 2 [26].

The solubility of CNT, CNT-Br, and CNT-NTf_2_ in water and ethanol is depicted in Figure 3. Generally, agglomeration of CNTs occurred due to their large surface energies and high specific surface areas. After sonication, CNTs were deposited in water at the bottom of the container (Figure 3a). CNT-Br is soluble in water and ethanol between functionalized carbon nanotubes, but CNT-NTF_2_ is only soluble in ethanol and is insoluble in water. Thus, the functionalized CNTs’ solubility can be altered by simply transferring the anion (Figure 3b,c).

Figure 4 shows the results of the thermal properties investigation of functionalized CNTs by TGA and DTG. The pristine CNT was stable below ~600 °C. However, the butylimidazolium salt-grafted CNTs (CNT-Br and CNT-NTf_2_) started to decompose at a lower temperature. The functionalized CNTs represent to main regions of weight loss. The splitting-off of surface-attached IL was the reason for the first stage of decomposition (~300 °C for CNT-Br and ~400 °C for CNT-NTf_2_). The second stage (~600 °C) was attributed to the decomposition of CNT. By comparing the resulting curves of TGA for CNT-Br and CNT-NTf_2_ it could be understood that types of pendent counter-anions affect the thermostability of CNT [26].

### 2.2. 3D Printing of Modified CNT Nanocomposites

Hen et al. investigated the resulting composite’s mechanical properties on improving the mechanical properties of wholly thermoplastic composites (WTCs) applicable in FFF. Multiscale thermoplastic composites were prepared by combining supercritical carbon dioxide (scCO_2_)-aided exfoliation and dual-extrusion technologies. These composites were polypropylene (PP) reinforced with thermotropic liquid crystalline polymer (TLCP) and CNTs. They also employed maleic anhydride-grafted-polypropylene (MAPP), which acts as a compatibilizer for WTCs to enhance the mechanical properties. The maleic anhydride interacts with the amide groups in PP and, by hydrogen bonding, with ester groups in TLCP [28].

Melt blending of thermoplastic and exfoliated CNTs in a single-screw extruder was the method of the nanocomposite matrices preparation, and Vectra B950 pellets were the type of employed TLCP [29,30]. To obtain 1 wt% CNTs reinforced multiscale WTCs with 20 wt% and 30 wt% TLCP, 1.25 wt% CNTs/98.75 wt%, and 1.43 wt% CNTs/98.57 wt% thermoplastic polymer matrices were prepared, respectively. The TLCP and matrices were plasticized and extruded by a single-screw extrusion. The temperature of the static mixers and the die were kept at 270 °C and 250 °C, respectively, within a 3-mm-diameter die. Thermoplastics reinforced with CNT and TLCPs’ WTC filaments for FFF produced by the dual-extrusion process to print specimens for tensile test [28].

Tensile test results revealed that by adding 1 wt% CNTs to the base composite system (20 wt%TLCP/PP), the tensile strength improved remarkably compared to adding 16 wt% MAPP. The combination of CNT and MAPP showed the maximum tensile strength. Therefore, the tensile strength increased to 128.83 (43.8%), 106.37 (42.7%), and 89.59 (20%) MPa for the additions of 1 wt% CNT + 16 wt% MAPP, 1 wt% CNT, and 16 wt% MAPP, respectively, to the base composite (Figure 5) [28]. The SEM images of fractured surfaces in Figure 6 highlight the strong adhesion among the TLCP fibrils, the polymeric matrix, and the CNTs.

After 3D printing of specimens via FFF with a rectangular shape, all the printed samples in the concentric pattern excluding 20 wt% TLCP/PP, had a smooth surface with no warpage. The reason is that CNTs and/or MAPP aided the high alignment of advanced filaments. The WTC samples were printed in rectilinear pattern (samples with no CNT), and both had warpage. In contrast, multiscale WTC (samples with CNT) did not represent warpage. This phenomenon could be attributed to the higher thermal conductivity of CNT reinforced WTCs compared to normal WTCs with no CNT. The blending of polymer with high thermal conductivity materials causes a reduction of thermal gradient in FFF. Subsequently, it leads to least the warpage [28,31].

Figure 7 represents the tensile strength and modulus of 3D printed WTC and multiscale WTC filaments. A meaningful improvement in all multiscale WTC compared to WTC is evident. By adding 1 wt% CNTs, WTC based on PP and MAPP/PP printed in the concentric pattern, the tensile moduli were 11.23 GPa and 18.98 GPa, increasing by 127.8% and 211.1%, respectively. At the same time, the tensile strength increased by 34.2% and 27.6% to 53.67 MPa and 61.27 MPa, respectively. The tensile strength and modulus of fused fabricated 20 wt% TLCP/1 wt% CNTs/16 wt% MAPP/PP was comparable with similar research on 40 wt% SCF/PP, 48.5 wt% SGF/PP, and 9 wt% LCF/nylon [32].

As mentioned earlier, one of the drawbacks of CNTs is their poor dispersion and solubility in a composite. In addition, the strong van der Waals forces result in cohesive interactions [33], which limits their excellent properties. Ultrasonic dispersion [34], surface treatment by surfactants [35], chemical modifications [36], and grafting improve the CNTs-polymer matrices compatibility [26]. Lately, filled polymers with electrically conductive nanowires or particles have been 3D printed to produce electronic devices on several substrates [37]. In addition, utilizing the 3D printing approach to design pressure sensors is another application in recent articles [38]. The FFF printing technique is an efficient method to print any composite-based physical sensors due to various advantages. This technology allows the fabrication of even slight indentation on the final printed parts by employing support from lower layers [39].

Kin et al. fabricated a filament of PLA/CNT and used it in a 3D printer as a part of their study on the pressure sensors. MWCNTs/PLA filaments were obtained by first dispersing commercial MWCNTs’ powders in dichloromethane (DCM) using a probe-type ultrasonicator at 24.86 kHz and180 W for 5 min. Then, 50 g of dried PLA pellets were dissolved in 500 mL of the MWCNTs/DCM suspension by agitation at 600 rpm at room temperature for 8 h. The evaporation of DCM solvent occurred at 100 °C for 12 h. Finally, the dried MWCNTs/PLA composite was shredded and then extruded through a single screw extruder at 180 °C and 45 cm/min speed to fabricate the MWCNTs/PLA filament (Figure 8) [39].

Electrical and thermal property characterization of MWCNTs/PLA composites performed on the filament (before printing), and 3D printed samples with different weight percentages (wt%) between 0% to 4%. First, the filament specimen was cut into a length of 55 mm. Next, it was coated with conductive epoxy at both ends and then heated at 60 ◦C in a convection oven for 4 h to cure the conductive epoxy. Next, the 3D printed specimen was fused fabricated in 10 × 10 × 3 mm by a dual nozzle layer height: 0.2 mm, printing speed: 45 mm/s, line width: 0.5 mm). Again, the conductive epoxy was coated on both sides of the surfaces by attaching the copper tape to create electrical contacts. Finally, to determine mechanical properties, the filament (before printing) was cut with a length of 70 mm. Then, the 3D specimen was printed out for the tensile test based on the ASTM D638 Type V specimen (Length: 9.53 mm, Width: 3.18 mm, and thickness: 3.2 mm) [39].

By applying the following equation (Equations (1) and (2)), the conductivity of each sample was calculated:(1)$$\sigma_{Filament}=\frac{1}{R_{Filament}}\times \frac{l}{\pi \times {\left(\frac{D}{2}\right)}^{2}}$$
(2)$$\sigma_{3D-printed}=\frac{1}{R_{3D-printed}}\times \frac{H}{A}$$
where: *l* is the length between the coated conductive epoxy and *D* is the filament diameter; *H* is height, and *A* is the area of coated conductive epoxy of the 3D printed sample [39].

Concerning the composite filament, the electrical conductivity enhanced dramatically to 0.5 wt% of filler contents. An entire percolation network of MWCNTs is revealed at more than 1 wt% of filler contents. Therefore, the results showed that the MWCNTs/PLA composite is stable based on recent reports of percolation theory [40]. The conductive networks were well-formed between the PLA matrices when the composite system contained more than 1 wt% MWCNT fillers. Although similar to the filament, the electrical conductivity case of the 3D printed sample increased with the enhancement of MWCNTs. However, the electrical conductivity in the 3D printed part is lower than that of filaments. It could be attributed to the created voids (intravoids) [5] between the layers of the 3D printed part and an incomplete layer of filament as a function of temperature.

The MWCNT has a negative temperature coefficient of resistance (TCR), representing the semiconductor essence of MWCNTs [41]. In the filament, increasing temperature up to 75 °C decreased the resistance, and the resistance increase after 75 °C was due to thermal expansion of the polymer matrix (Figure 8c). However, in the 3D printed specimen, the resistance began to raise from 40 °C. This trend may be due to the thermal stress that developed during the 3D printing process in the specimen and accelerated the thermal expansion at a lower temperature compared to the filament specimen. Hence, there is an inconsistency in thermal behaviors between the filament and the 3D printed part [39].

Concerning the mechanical properties, the whole composite system (polymer + filler) is reinforced by the MWCNT. Based on Figure 9a, which represents Young’s modulus of the filament and the 3D printed part, the addition of 4 wt% MWCNT enhanced Young’s modulus by 41.57% and 59.34% for the filament and the 3D printed part, respectively. It could be due to the transfer and distribution of the externally applied stress along the MWCNT to the composite in the polymer matrix. Figure 9b,c reveals the stress-strain curves of the 3 wt% loaded MWCNT in the filament and 3D printed part. In the case of filament, necking and fracture happened at 6% and 15% strain, respectively. While the 3D printed specimen fractured at 8% strain with no sign of necking and showed a lower Young’s modulus than the filament. Figure 9d,g shows the digital image of the filament and 3D printed part. Figure 9e–i shows the surface morphology of the produced parts. Opposite to filament, 3D printed samples experience incomplete interfacial bonding and voids between the layers. These voids and interlayers in 3D printed specimens caused a reduction in Young’s modulus. Although it is represented that the enhancement of MWCNT wt% could tune the mechanical properties of the 3D printed part [39].

The 3 wt% of MWCNTs/PLA was selected for 3D printing to achieve higher electrical conductivity and appropriate mechanical properties. Figure 9h,i shows the presence of voids in the wider raster of the printed part. The difference in the temperature of previously deposited raster and later deposited raster in the process of melting and solidifying can cause weakening the bonding between the printed layers, leading to the brittle fracture of 3D printed composites [39,42].

Unfortunately, current techniques of 3D printing are not appropriate for significant polymer structures with geometrical characteristics at the sub-micrometer scale [43]. Furthermore, almost every 3D orienting technique is subjected to a trade-off between printing time, building volume, and the printing voxel [44].

As a result, polymer objects with 1–100 nm nanoporosity are currently out of reach [45]. Instead, recent studies have recommended a straightforward 3D printing method based on phase separation of liquid resins to produce complex geometry glasses with higher resolution and multi-oxide chemical compositions by employing a desktop light processing (DLP) printer [45]. Based on what has been expressed so far as well as relying on recent studies, utilizing CNTs and other fillers has improved the properties of the final composite system, but it has remained some potential to be increased. For this purpose, application of the phase inversion process for composite formation could be a viable option. The mechanism and types of phase separation have explained in detail in the introduction section of this paper. In addition, an example of phase inversion application in composite formation has been mentioned to highlight the concept.

## 3. An Alternative to Improve Nanocomposite Properties

We propose a new approach that could overcome the limitations to acquire a well dispersed and high-performance nanocomposite by considering the constraints above. The process is under investigation and implies the phase inversion technique. Dilute polymer/good solvent/CNT suspensions will be conveyed to an aqueous solution to remove the solvent and extract the polymer/CNT system. This method has the advantage of obtaining a well-dispersed polymer/CNT mixture.

### 3.1. Phase Inversion Technique

Phase inversion is a process in which a homogeneous polymer solution undergoes a phase transformation in a controlled way from liquid to a solid state. There are four different methods to perform this phase transformation, namely [46,47]:(a)Immersion precipitation. A polymer solution is soaked in nonsolvent coagulation or precipitation bath (commonly water) in this method. The polymer coagulation occurs through the desorption of the solvent in the precipitation bath.(b)Thermally-induced phase separation. This technique relies on diminishing the solvent’s efficiency by temperature reduction. Subsequently, the solvent is removed by freeze-drying, evaporation, or extraction.(c)Evaporation-induced phase separation. The polymer solution is produced in a solvent or a combination of volatile nonsolvent, and the evaporation of the solvent results in precipitation or demixing/precipitation. This method is also called solution casting.(d)Vapor-induced phase separation. The polymer solution is exposed to an atmosphere containing nonsolvent vapor (usually water). Therefore, demixing/precipitation occurs by the absorption of nonsolvent.

The first two methods produce polymeric membranes with disparate morphologies [46,47,48]. In Figure 10, the schematic of polymeric membranes obtained by using the phase-inversion techniques is reported. In general, asymmetrical polymer films with thin, relatively dense skin and porous substrate are obtained. The step of soaking and immersion is generally called the quenching step. Nonsolvents are usually compatible with the solvents but do not dissolve the polymer constituents. Therefore, the final properties of the membrane are dependent on variables such as the initial polymer concentration in the polymer casting solution, initial solvent concentration in the coagulation bath, relative amounts of polymer and nonsolvent in casting solutions, etc. On this basis, numerical models to describe membrane formation have a considerable value in determining the properties of the resulting membrane [49,50,51].

The mass exchange of solvents and nonsolvents through the casting solution/coagulation bath interface results in the precipitation of polymer constituents. Subsequently, two disparate phases of polymer-rich and diluted polymer are obtained, the former being referred to as membrane. Crucial information about the structure and substructure of the produced membrane could be achieved by modeling mass transfer phenomena. Elementary assumptions based on Fick equations are considered for this subject despite its complexity. A fair agreement with the available experimental data was obtained with material parameters assumed to be constant in two polymeric systems investigated in [50]. Furthermore, the mass transfer process dynamics appeared to be much more important than the influence of the concentration on the diffusion and thermodynamic partition coefficients.

According to the assumptions above, the mass transfer of component *i* within the phase *j* can then be written as:(3)$$\frac{\partial C_{\left(i,j\right)}}{\partial t}=D_{\left(i,j\right)}\times \frac{\partial^{2}C_{\left(i,j\right)}}{\partial x^{2}},\;\{\begin{array}{ll} i=1,2, & for\;j=1\\ i=1, & for\;j=2 \end{array}$$
where, *C*(*i*,*j*) is the molar concentration of component *i* in phase *j* (kg/m^3^), *D*(*i*,*j*) is the diffusion coefficient of component *i* in phase *j* (m^2^/s), and *x* is the mass transfer direction (m). This is subject to the following boundary conditions:(4)$$\frac{\partial C_{\left(i,1\right)}}{\partial x}{|}_{x=0}=0,\;i=1,2,\;\;for\;j=1$$
(5)$$\frac{\partial C_{\left(i,2\right)}}{\partial x}{|}_{x=w}=0,\;i=1,\;\;for\;j=2$$
(6)$$C_{\left(i,1\right)}{|}_{L}=k_{\left(i\right)}\times C_{\left(i,2\right)}{|}_{L},\;\;i=1,2$$
(7)$$D_{\left(i,1\right)}\;\frac{\partial C_{\left(i,1\right)}}{\partial x}{|}_{x=L}=D_{\left(i,2\right)}\times \frac{\partial C_{\left(i,2\right)}}{\partial x}{|}_{x=L},\;\;i=1,2$$
where *L* is the casting solution thickness at time t(m), *k(i)* is the partition coefficient of component *i*, and W is the external limit of the coagulation bath. Equations (4) and (5) indicate no mass exchange through the system’s borders. Equation (6) describes the local thermodynamic equilibrium at the interface. Equation (7) describes the mass transfer continuity at both sides. Assuming that the initial solutions are homogeneous, it is possible to write:(8)$$C_{\left(i,j\right)}{|}_{t=0}=C_{0\left(i,j\right)},\;\;i,\;j=1,2$$

The above equations establish the mathematical model applied to simulate the phase inversion phenomena. To solve the model, fundamentally needs to know the molecular weights, densities, diffusion coefficients, and partition coefficients for all the chemical species involved [50].

### 3.2. Phase Inversion for CNT Nanocomposite Fabrication

Accordingly, the scheme of the proposed technique for manufacturing nanocomposite filaments for 3D printers is shown in Figure 11. Following a homogeneous CNT/polymer solution immersion in the coagulation bath, the solvent is removed from the polymer solution due to its affinity with the nonsolvent. As a result, the polymer recovers its solid state and remains pervaded by the dispersion of the CNTs.

During the phase inversion step, an external force is required to pull out the precipitated composite polymer, which now is a kind of membrane and is almost glassy and brittle. The process is designed to stretch the filaments to increase the CNTs’ alignment and obtain a near filament shape for a 3D printer. However, the filament shape cannot change after phase inversion. Thus, the stretching process is conceived at the same time with phase inversion occurrence. For this purpose, considering the volume consistency rule, altering the velocity of the filament (at the diameter of *d*_2_) could solve the issue. Based on volume consistency rule:(9)$$V_{1}=V_{2}$$
(10)$$v_{1}A_{1}=v_{2}A_{2}$$
where, *V*_1_, *V*_2_ are the volume of input and output material to a die during phase inversion, and v and A are the velocity and surface area, respectively, before (zone 1) and after (zone 2) shape deformations. By a simplification, it results in the Equation (11):(11)$${v_{1}d_{1}}^{2}={v_{2}d_{2}}^{2}$$
where *d*_1_ and *d*_2_ are the diameter of melt polymer and glassy polymer, respectively. Hence, reduction in the diameter results from filament stretching and alignment of CNTs reaching a favorable diameter for the filaments.

Lannoy et al. [52] investigated the carboxylic functionalization of CNTs on polysulfone (PSf) ultrafiltration membranes. The mixture of CNTs and DMF, chilled in an ice bath to 8 °C, and probe ultra-sonicated with intervals of 3 s for a total of 1 h at a power output of 70 W. Once achieved a suspension and addition of PVP to that, all the systems agitated for 30 min. Then, by heating to 60 °C and stirring steadily, a homogeneous mixture was obtained in which the PSf is dissolved. Using a casting knife, the final solution was then cast and set to a distance of 250 μm away from a glass plate. The cast solution was let to stabilize and de-gas on the glass plate before plunging in a specified volume of DIW. The membrane precipitated and detached from the glass plate. The final step was staying the membrane in the bath for 20 min, transferring it to fresh DIW, and storing it at 4 °C for 24 h [52].

MWCNTs were mixed with a ratio of sulfuric to nitric acid of 3:1 to create a quite oxidizing environment. Thus, the control of the functionalization degree of CNTs was feasible by altering either the concentration of acids or the reaction time represented in Table 1.

CNTs were first sonicated in DMF to achieve a homogeneous dispersion where the polymer was dissolved. Stable carboxylated CNTs suspension was obtained after sonication. In contrast, pristine CNTs participated out of suspension in few hours, as expected. After sonication, PVP and PSf were added to the solvent–CNT suspension. The final mixture, colored black, was homogenous. The produced viscous solution was cast on glass and participated in DIW. The initial polymer nanocomposite solution consisted of 0.5 wt% CNTs concerning the polymer constituents for all the membranes. SEM images of produced membranes are depicted in Figure 12a,b. The cross-section images showed that membranes created with pCNTs consisted of broad cumulative found predominantly within them. However, individual CNT fibers were rare. Membranes produced with CNT-COOHs of all functionalization weight percentages represented more uniform dispersion. The individual CNTs inside the bulk were recognizable, and there were no cumulative CNTs [52].

Several phenomena were observed during immersion precipitation of the PSf–PVP–MWCNT solution: (1) with precipitation of polymer system, small black plumes were perceived arises from the membrane, (2) these black plumes were more visible for membranes that were composed of CNTs with a higher degree of carboxylation, (3) it was noticed that the active side of the membrane is darker than the support side (Figure 12c), (4) as represented in Figure 12d the membrane consisted of CNTs with a higher degree of carboxylation were usually lighter in color than those with a lower degree of functionalization [53].

The precipitation immersion led to enormous changes in the polymer nanocomposite system [53]. The functionalized CNTs were dissolved in DMF, an organic solvent, while they were less compatible with PSf. In addition, exploiting water as a nonsolvent presented a favored phase for CNT-COOHs. By the diffusion of DMF through the polymer solution and precipitation, the CNTs migrate approaching the side of the membrane that was in contact with water, i.e., the active side (Figure 12c). The CNTs’ transfer to the active side was followed by a getaway into the nonsolvent bath and highlighted by the color conversion in the membrane (Figure 12d) and the black plumes that arose in the nonsolvent bath during precipitation immersion [52].

To calculate the number of losses in CNTs, UV–vis absorption of the post-precipitation nonsolvent bath was measured for each type of membrane (Figure 13). It was represented that CNTs with a higher degree of carboxylation were more vulnerable to loss from the precipitating membrane, due to affinity for the polar nonsolvent. Conversely, membranes with a lower degree of carboxylation experienced better resistance to the CNT loss from their polymer matrices [52].

Figure 14 demonstrates an unusual trend of Young’s modulus change by altering the degree of carboxylation of CNTs within the membrane. Membranes composed of pCNTs and CNT-COOHs revealed higher Young’s moduli than pure polymer membranes, however, for CNT carboxylation as low as 2.56%, the membranes formed with solo CNT-COOHs showed enhancement in Young’s modulus over that of pCNT. Hence, at higher degrees of carboxylation of CNTs, there was a reduction in the physical strain properties of membranes compared to that formed with pCNTs. pCNTs only caused an improvement in tensile strength in localized zones with weak bond bearing. They led to limited increases in Young’s modulus. While carboxylated CNTs improved the homogeneous distribution of CNTs into polymer solutions and enhanced the membrane’s load-bearing [52].

Although, this enhancement in the load-bearing by improvement of homogeneity was in balance with s tensile strength reduction of the individual CNTs and fewer CNTs existence in the membrane. Young’s modulus as a function of the CNTs loss from the membrane is represented in Figure 14. However, the CNTs’ distribution all over the membrane was non-uniform (Figure 12). This non-uniform distribution reduces the potential of bulk load-bearing considerably. Moreover, this was connected to the decrease in strength of each CNT. The Young’s modulus of the membranes was shortened by decreasing the CNTs due to the increase in carboxylation [52].

## 4. Conclusions

Considerable progress has been achieved in AM and the application of CNT reinforced filaments over the last decades. CNTs exhibit exceptional mechanical, thermal, and electrical properties. However, their full potential has not been achieved when mixed with polymer matrices in a nanocomposite system. This study provides a survey of the research in carbon nanotube-polymer composites, focusing on the CNTs’ fundamentals of the functionalization methods, grafting, and composite formation. Agglomeration phenomena almost suppress the CNTs’ effectiveness due to the strong interaction force between CNTs.

The efforts made so far to increase the efficiency of the CNT-based reinforcement show that the different processes and different functionalization procedures have not obtained the expected results. The low CNT amount prevents the filament from strengthening from one side while the CNT flexibility reduces their reinforcing efficiency from the other side.

The phase-separation method mentioned herein may partly alleviate the shortcomings. First, it may reduce the CNTs’ agglomeration by dispersing the nanotubes in a dilute solution. Secondly, the CNTs can be oriented along the direction of the filament by a suitably stretching process. The mechanism and basic equations governing the phase inversion have been discussed, and a preliminary approach for high-performance nanocomposite fabrication was suggested. To produce high-strength nanocomposite filament used in 3D printers, the proposed method’s effectiveness will be fully explored in the following paper.

## Figures and Tables

**Figure 1 polymers-13-04007-f001:**
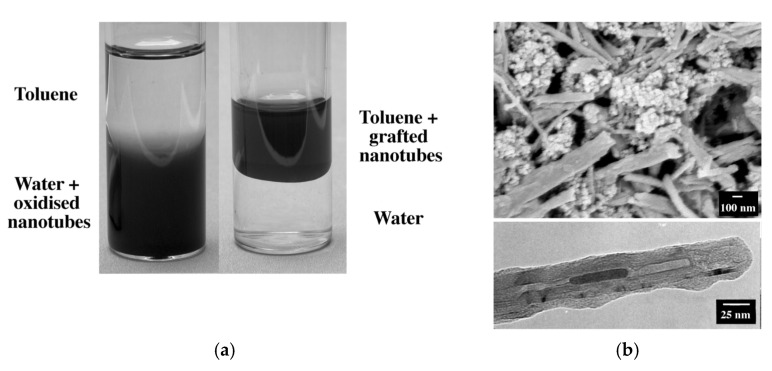
(**a**) Solubility of nanotubes in different environments. (**b**) SEM (top) of partially purified grafted nanotubes (the polymer aggregates disappeared after further purification) and TEM (bottom) results of a single grafted nanotube. With permission [24].

**Figure 2 polymers-13-04007-f002:**
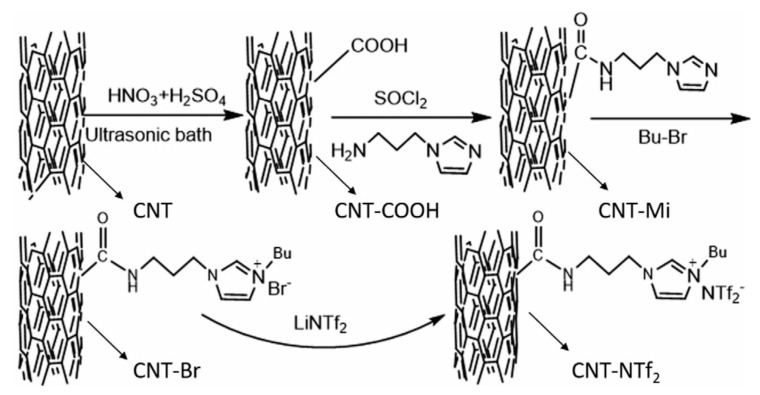
Synthetic route of ionic liquids functionalized CNTs. With permission [26].

**Figure 3 polymers-13-04007-f003:**
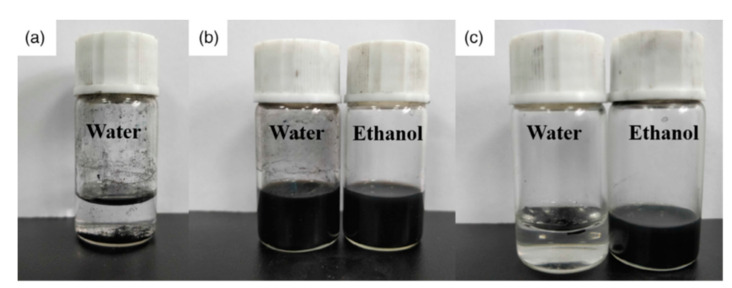
The solubility of CNT (**a**) in water and CNT-Br (**b**) in water and ethanol and CNT-NTf_2_ (**c**) in water and ethanol, from left to right with contents of 5 mg in 10 mL solutions. With permission [26].

**Figure 4 polymers-13-04007-f004:**
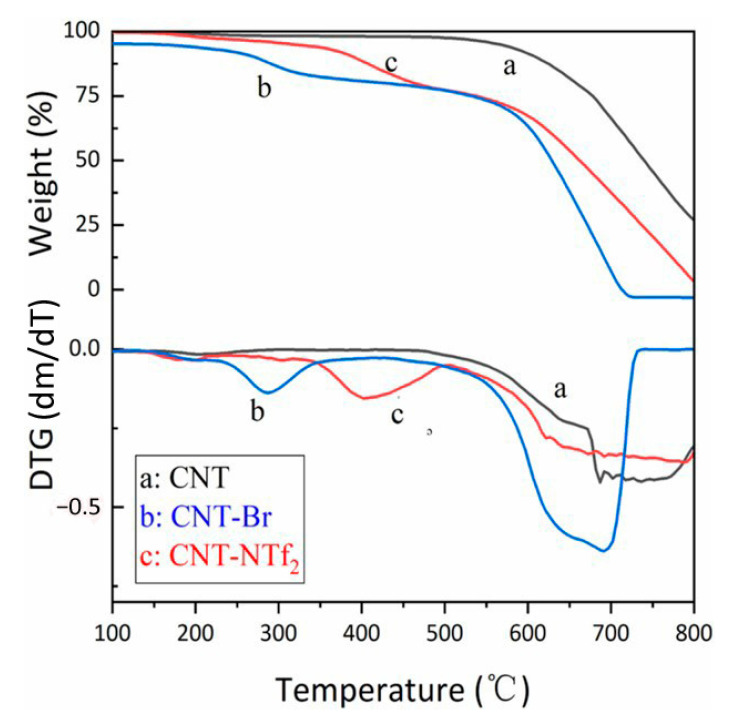
TGA and DTG of CNT (**a**), CNT-Br (**b**), and CNT-NTf_2_ (**c**). TGA is the thermogravimetric analysis and DTG is the first derivative of TG in which m and T are indicators of mass loss and temperature, respectively. With permission [26].

**Figure 5 polymers-13-04007-f005:**
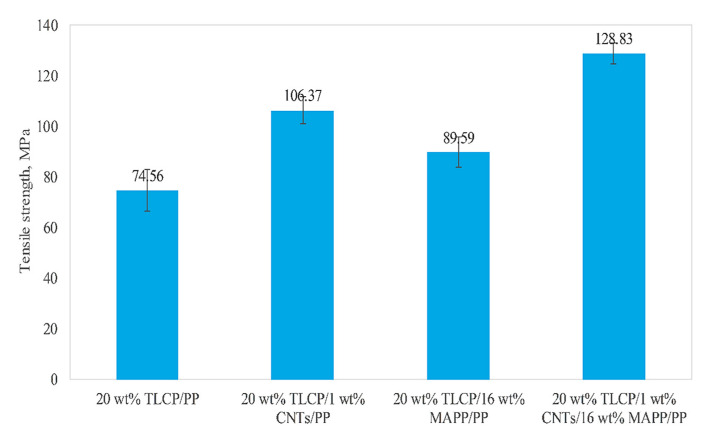
Tensile strength of 20 wt% TLCP reinforced WTC and multiscale WTC based on PP in the form of filament strands. With permission [28].

**Figure 6 polymers-13-04007-f006:**
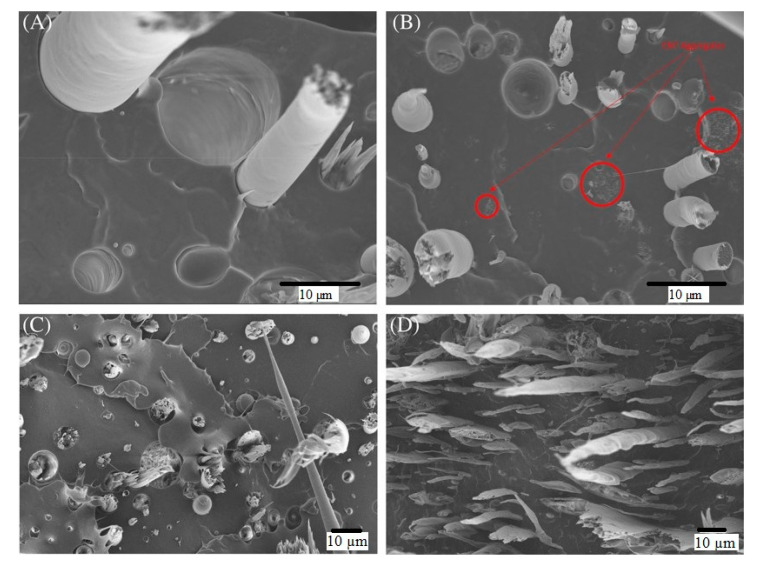
SEM micrographs of fracture surface of filaments of (**A**) 20 wt% TLCP/PP; (**B**) 20 wt% TLCP/1 wt% CNTs/PP; (**C**) 20 wt% TLCP/16 wt% MAPP/PP; (**D**) 20 wt% TLCP/1 wt% CNTs/16 wt% MAPP/PP. With permission [28].

**Figure 7 polymers-13-04007-f007:**
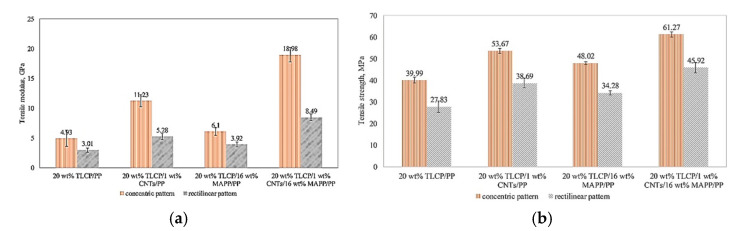
(**a**) Tensile modulus and (**b**) Tensile strength of laid down specimens printed in concentric and +45° rectilinear printing patterns. With permission [28].

**Figure 8 polymers-13-04007-f008:**
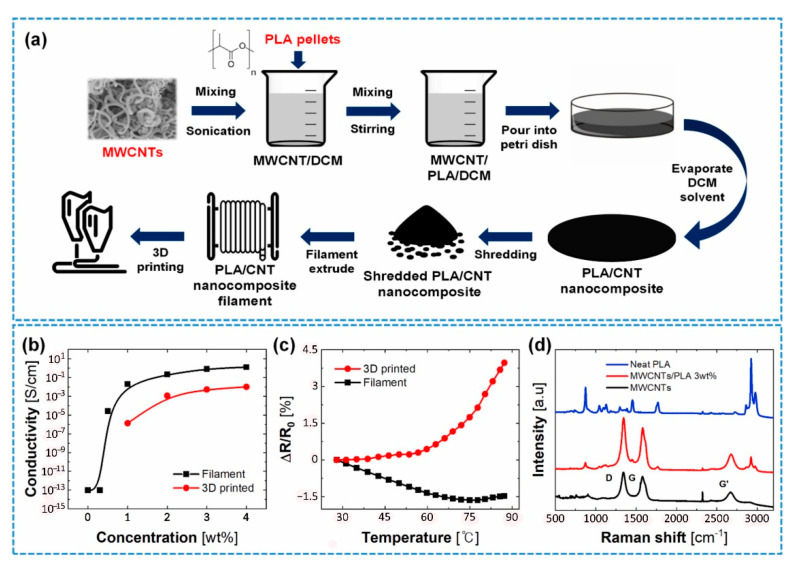
(**a**) Schematic process of the composite filament production. (**b**) Electrical conductivity of the filament and 3D printed sample of MWCNT composite. (**c**) Relative resistance changes-temperature curve of the 3 wt% filaments and 3D printed sample. (**d**) Raman spectroscopy of pristine PLA, MWCNT, and MWCNTs/PLA composite. With permission [39].

**Figure 9 polymers-13-04007-f009:**
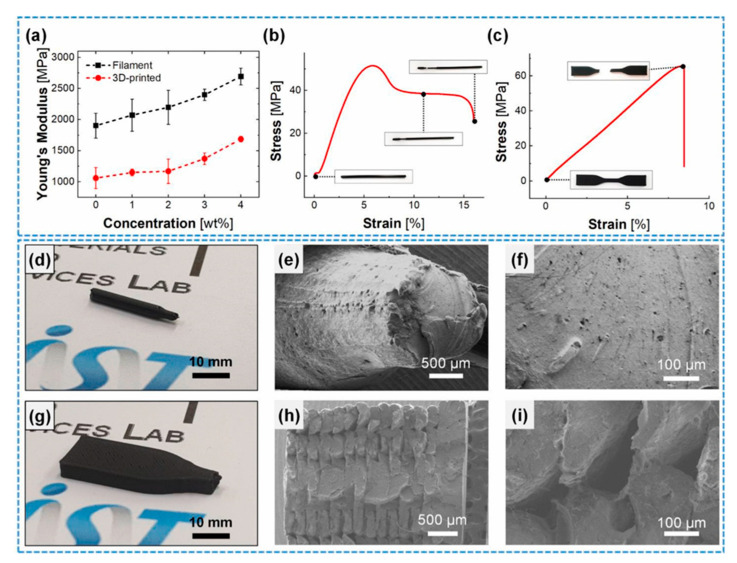
(**a**) Young’s modulus with various concentrations of MWCNT fillers. Stress-strain curve of 3 wt% (**b**) filament, (**c**) 3D printed specimen. Optical and SEM images of the fracture surface of (**d**–**f**) 3D printed specimen, (**g**–**i**) filament. With permission [39].

**Figure 10 polymers-13-04007-f010:**
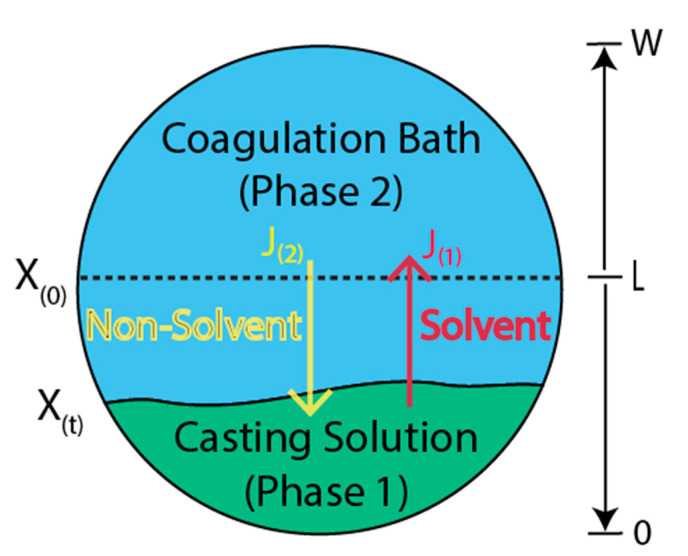
Schematic representation of the phase-inversion process. Adapted from [50].

**Figure 11 polymers-13-04007-f011:**
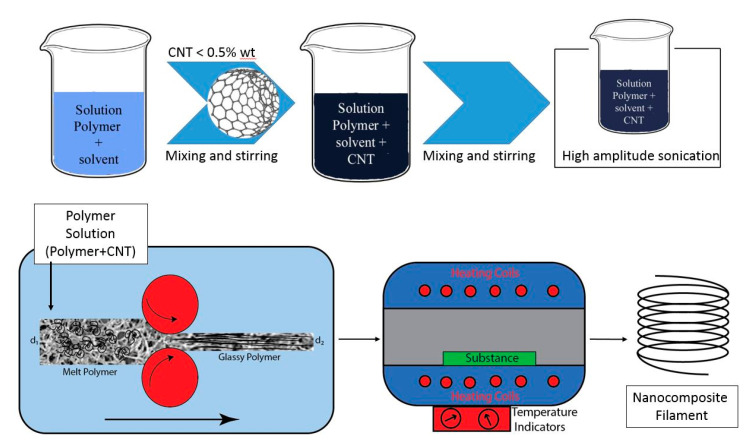
Schematic of suggested approach for filament fabrication.

**Figure 12 polymers-13-04007-f012:**
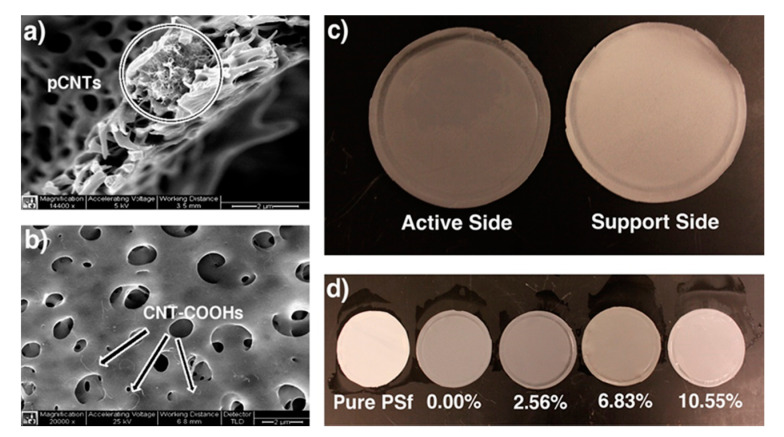
(**a**) Cumulated pristine MWCNTs inside the cross-section of a PSf membrane. (**b**) Well-dispersed and functionalized CNT-COOHs (7.97%) all over the polymer membrane. (**c**) The darker color of the active side compared to that of the support side. (**d**) The membrane color is an indicator of the compatibility of PSf with CNTs in different ratios of carboxylation (on left only PVP and PSf, others are nanocomposites with 0.5 wt% CNTs). With permission [52].

**Figure 13 polymers-13-04007-f013:**
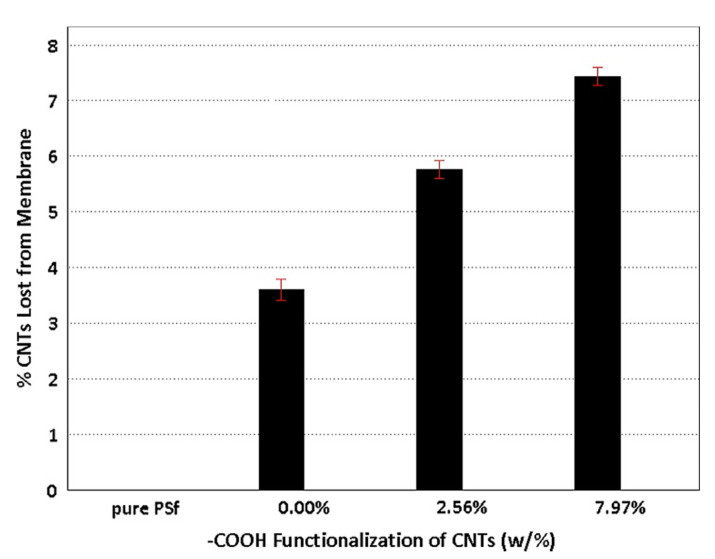
CNTs loss from polymer membranes during precipitation immersion in DIW. With permission [52].

**Figure 14 polymers-13-04007-f014:**
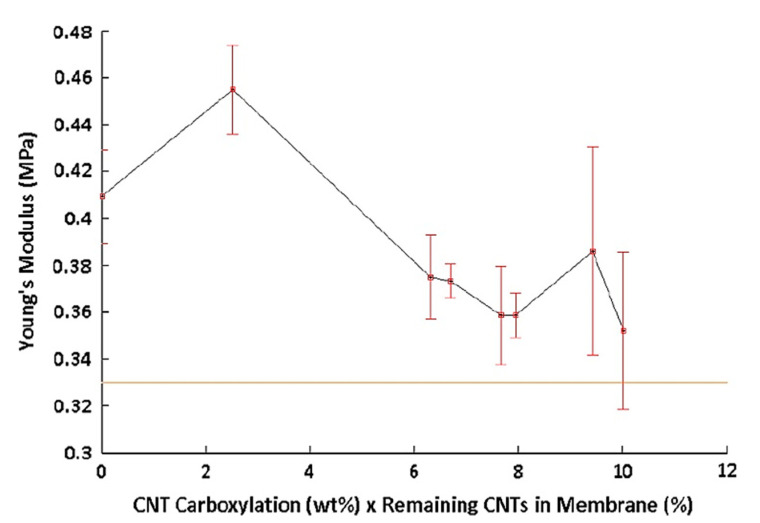
Young’s modulus vs. PSf membranes containing CNTs with different degrees of carboxylation. PSf membranes containing 0.5 wt% CNTs of all carboxylation showed improved Young’s Moduli over pure PSf membranes (orange line). With permission [52].

**Table 1 polymers-13-04007-t001:** The reaction circumstances were applied to obtain different degrees of carboxylation of MWCNTs [52].

Carboxylation Degree (wt%)	0	6.53	6.94	7.97	8.29	9.91	10.55
Reaction time (h)	0	1	2	4	4	8	8
Volume of acids to mass of CNTs (mL/g)	0	150	150	75	150	150	75

## Data Availability

The data presented in this study are available on request from the corresponding author.

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
