# Peer review of "Fabrication of High-Performance CNT Reinforced Polymer Composite for Additive Manufacturing by Phase Inversion Technique"

_polymers, 2021, doi:10.3390/polym13224007_

Round 1
Reviewer 1 Report
The paper is focused on the fabrication of nanocomposites based on polymer and CNT. The topic is interesting for readers of Polymers. I recommend the publication after the following minor revisions:
- Some typos and grammar mistakes are present in the manuscript. For instance, in the Abstract (line 19) it is reported “discuses” instead of “discussed”. Please check and correct.
- In the Introduction, I suggest to quote other kind of hollow tubular nanostructures (halloysite nanotubes, boron nanotubes, imogolite nanotubes) that can be used as reinforcing fillers for polymers. This consideration could be supported by recent reviews [DOI: 10.3390/app8071068; DOI 10.1016/j.carbpol.2020.116502].
- The scale length within SEM images (Fig. 5) is not clear. Please check and revise.
- 9. Please report the unit for the y-axis of the DTG curves.
- 10. The equations should be numbered.
- Conclusions paragraph should be extended.
Author Response
Response to Reviewer 1
- Comment 1: The paper is focused on the fabrication of nanocomposites based on polymer and CNT. The topic is interesting for readers of Polymers. I recommend the publication after the following minor revisions.
Response: Thank you very much.
- Comment 2: Some typos and grammar mistakes are present in the manuscript. For instance, in the Abstract (line 19) it is reported “discuses” instead of “discussed”. Please check and correct.
Response: Thank you for pointing this out. We have made the revision accordingly
- Comment 3: In the Introduction, I suggest to quote other kind of hollow tubular nanostructures (halloysite nanotubes, boron nanotubes, imogolite nanotubes) that can be used as reinforcing fillers for polymers. This consideration could be supported by recent reviews [DOI: 10.3390/app8071068; DOI 10.1016/j.carbpol.2020.116502].
Response: Thank you for this suggestion. The proposed papers have been reviewed completely. They have reported some substitution for CNTs which is a key factor in our current paper. In addition, the ability to improve in the mechanical, thermal, and electrical properties of composites after adding CNT fillers is an advantage that suggested fillers don’t have. Although the second suggested paper focused on the fillers’ biodegradability, both references have been indicated in the revised version.
- Comment 4: The scale length within SEM images (Fig. 5) is not clear. Please check and revise.
Response: Thank you for pointing out this. All the images again have been checked and replaced with a higher quality in case of available items. The scale bar of Figure 5 has changed totally.
- Comment 5: Please report the unit for the y-axis of the DTG curves.
Response: Thank you for your kind reminder. Theoretically, DTG is the derivation of the TG, so this units would be (dm/dT), which has been noted in Figure 9 and expressed in its caption.
- Comment 6: The equations should be numbered.
Response: Thank you for your kind reminder. The equations have been numbered in the revised version.
- Comment 7: Conclusions paragraph should be extended
Response: Thank you so much. The conclusion has been developed by mentioning other essential outcomes of the revised paper.
Reviewer 2 Report
This paper is a communication proposing based on the phase inversion method, a novel technique to produce CNT-reinforced filaments to simultaneously increase the mechanical, thermal, and electrical properties. Indeed is an interesting paper having a novel approach for fabrication of CNT reinforced polymer composite with high performance for additive manufacturing.
The functionalization procedures for the preparation of CNT reinforced composites are discuses in this study based on well selected references. Such comments sustained the merits of the article which is also clearly written and deserves publication in journal Polymers which is a good option as well
The main demerit of the present paper is organization which in my opinion needs to be seriously revisited. Firstly the number of subchapters are not enough without conclusions and introduction being only one subchapter entitled „Polymer/CNT composites” Schematic of suggested approach for filament fabrication. The introduction has a schematic representation of the phase-inversion process. and a schema of suggested approach for filament fabrication. May be it will be in the benefit of article such part to be represented in a separate subchapter. In such way the subchapter about polymer/CNT composites could be divided in two or even three parts as well.
The fact that the paper has only 1 table and 14 figures will be better to be reorganized taking into account that the majority of figures are taken with permission from various other paper . It is written that this fact was done with permission, but only mentioning such things is not in the benefit of paper novel character
Author Response
Response to Reviewer 2
- Comment 1: This paper is a communication proposing based on the phase inversion method, a novel technique to produce CNT-reinforced filaments to simultaneously increase the mechanical, thermal, and electrical properties. Indeed is an interesting paper having a novel approach for fabrication of CNT reinforced polymer composite with high performance for additive manufacturing.
The functionalization procedures for the preparation of CNT reinforced composites are discuses in this study based on well selected references. Such comments sustained the merits of the article which is also clearly written and deserves publication in journal Polymers which is a good option as well
Response: Thank you very much.
- Comment 2: The main demerit of the present paper is organization which in my opinion needs to be seriously revisited. Firstly the number of subchapters are not enough without conclusions and introduction being only one subchapter entitled „Polymer/CNT composites” Schematic of suggested approach for filament fabrication. The introduction has a schematic representation of the phase-inversion process. and a schema of suggested approach for filament fabrication. May be it will be in the benefit of article such part to be represented in a separate subchapter. In such way the subchapter about polymer/CNT composites could be divided in two or even three parts as well.
Response: Thank you very much for your comments that helped us improve this manuscript. We agree with this comment and the paper has been revised completely to be organized better. In addition, we changed the order of contents. Finally, by adding more subtitles, we believe that the reading of this paper got easier based on your comment.
- Comment 3: The fact that the paper has only 1 table and 14 figures will be better to be reorganized taking into account that the majority of figures are taken with permission from various other paper . It is written that this fact was done with permission, but only mentioning such things is not in the benefit of paper novel character
Response: You have raised an important point here. This paper is indeed a communication paper. After addressing the problem by mentioning current research, we proposed the novel idea behind it. All the figures and the only table of this paper have the permission of the publishers. These permissions were released before submitting the article.